

# Correlation of the sperm DNA fragmentation index with semen parameters and its impact on fresh embryo transfer outcomes—a retrospective study

Xiao Ju Wan, Meizhen Huang, Min Yu, Tao Ding, Zhihui Huang, Zhiqin Zhang, Xingwu Wu and Jun Tan

Jiangxi Maternal and Child Health Hospital, Nanchang, China

## ABSTRACT

**Purpose:** To evaluate the relationships between the sperm DNA fragmentation index (DFI) and semen parameters and its application in fresh embryos *via in vitro* fertilization and transfer.

**Methods:** A retrospective propensity score matching (PSM) study was conducted on 162 cycles of *in vitro* fertilization (IVF) or intracytoplasmic sperm injection (ICSI) and fresh embryo transfer (fresh IVF/ICSI-ET) from 2020–2024.

**Results:** Spearman correlation analysis revealed a negative correlation between the sperm DFI and sperm motility ($r = -0.44$, $p = 3.32\text{e}{-}08$), forward movement of sperm ($r = -0.46$, $p = 3.25\text{e}{-}09$), and normal morphology of sperm ($r = -0.25$, $p = 0.000$); there was no correlation between the sperm DFI and male age ($r = 0.08$, $p = 0.31$), semen volume ($r = -0.15$, $p = 0.05$), sperm concentration ($r = -0.16$, $p = 0.32$), or male body mass index (BMI) ($r = 0.02$, $p = 0.98$). There were no significant differences in the 2PN fertilization rate (64.98% *vs.* 67.18%, $p = 0.362$), D3 high-quality embryo rate (28.34% *vs.* 23.91%, $p = 0.107$), biochemical pregnancy rate (71.60% *vs.* 71.60%, $p = 1$), clinical pregnancy rate (65.00% *vs.* 65.00%, $p = 1$), delivery rate (50.72% *vs.* 48.44%, $p = 0.928$), miscarriage rate (7.25% *vs.* 6.25%, $p = 1$), or singleton birth weight (3,350 g *vs.* 3,200 g, $p = 0.599$) between the normal DFI (DFI < 30%) group and the high DFI (DFI ≥ 30%) group.

**Conclusion:** Sperm DFI is weakly associated with sperm motility, forward movement, and normal sperm morphology, and the correlations are not strong. However, there is no correlation between the sperm DFI and the clinical outcomes of fresh IVF/ICSI-ET.

Corresponding author
Jun Tan, tanjun561127@163.com

Fifteen percent of couples of childbearing age worldwide have fertility issues, with male factors accounting for approximately 50% (*Agarwal et al., 2015*). Semen quality is an important indicator for evaluating male fertility. Conventional semen analysis mainly involves evaluating parameters such as the sperm concentration, motility, and morphology. However, traditional routine semen parameters can provide only limited

information and cannot fully reflect male fertility. On the one hand, approximately 15% of infertile men have normal semen parameters (*Agarwal & Allamaneni, 2005*). On the other hand, the reference values of routine semen parameters are not completely equivalent to those of fertility assessment and cannot be defined as the minimum value of male fertility (*Guzick et al., 2001*). Therefore, traditional semen parameter analysis has low predictive value for male fertility, and new biomarkers are needed to evaluate and predict male infertility.

In recent years, the sperm DNA fragmentation index (DFI) has been used to reflect the degree of sperm DNA damage and is considered a potential biomarker for auxiliary diagnosis and prediction of male fertility (*Campos et al., 2021*; *Zhang et al., 2021*). However, the results of evaluating the impact of increased DFI on outcomes in assisted reproduction technology (ART) are contradictory (*Maghraby et al., 2024*). Some studies have shown that an increase in the DFI has adverse effects on outcomes (*Li et al., 2024*), whereas other studies have not reported any impact (*Rios et al., 2021*; *Solanki et al., 2024*). One reason is that various methods have been used to detect the DFI in different published studies (*Rios et al., 2021*), and even with the same method, the laboratory thresholds and reagents used differ. Another reason is that ART uses sperm selection technology to eliminate low-quality sperm, which makes it difficult to interpret survey results. Here, we conducted a retrospective study with the following hypotheses: The primary hypothesis was that DFI is associated with ART outcomes. The secondary hypothesis was that DFI correlates with conventional semen parameters. This study aims to provide a clearer basis for utilizing DFI in the clinical assessment and prediction of male fertility.

## MATERIALS AND METHODS

### Study design

From November 18, 2020, to August 29, 2024, 2,317 patients who underwent *in vitro* fertilization (IVF) or intracytoplasmic sperm injection (ICSI) and fresh embryo transfer (fresh IVF/ICSI-ET) at the Jiangxi Maternal and Child Health Hospital, Reproductive Medicine Center, were retrospectively identified. The Reproductive Medicine Ethics Committee of Jiangxi Maternal and Child Health Hospital approved this study (SZYY-202407). The study was conducted in accordance with the Declaration of Helsinki. All participants provided written informed consent. During the research, we strictly followed the relevant guidelines and regulations of the local research institute. Participants were informed that they could withdraw from the experiment without giving a reason.

The inclusion criteria were as follows: (1) female infertility due to tubal factors and (2) fresh IVF/ICSI-ET with 1–2 transplanted embryos. (3) Oocyte retrieval time: 2020.11.18–2024.08.29. The exclusion criteria were as follows: (1) the female partner has polycystic ovary syndrome (PCOS), endometriosis, congenital uterine malformation or organic uterine lesions, *etc.*; (2) the male partner has azoospermia, retrograde ejaculation, and ejaculation disorders; (3) cycles with donor sperm and oocytes, and preimplantation genetic testing (PGT) cycle; and (4) missing data. DFI is classified as normal (DFI < 30%) or high (DFI ≥ 30%). A total of 1,110 infertile couples with fresh IVF/ICSI-ET were

included in this study, including 81 cases with high DFI levels and 1,029 cases with normal DFI levels. The study flowchart was presented in Fig. 1.

Considering that the sample size ratio between the normal DFI group and the high DFI group exceeded the range of 1:4, a 1:1 propensity score matching (PSM) method was used to match the female age, male body mass index (BMI), female BMI, female infertility duration, female anti-Mullerian hormone (AMH), fertilization method, controlled ovarian stimulation (COS) protocol, and number of embryos transferred from the grouped patients. Finally, 81 patients were included in the high DFI group, and 81 patients were included in the normal DFI group.

## Semen routine analysis

Semen processing and testing were performed according to the WHO Laboratory Manual for the Examination and Processing of Human Semen (6th edition) (*World Health Organization, 2021*). To ensure the accuracy of test results, internal quality control (IQC) measures were implemented throughout the testing process to minimize inter-technician variability. Additionally, the laboratory regularly participated in external quality assessment (EQA) programs (*e.g.*, the Sichuan-Chongqing Regional Quality Control Program in China).

The man abstained from sexual activity for 2–7 days and obtained semen through masturbation. The duration of abstinence was recorded. The weighing method was used to calculate semen volume (Ying Heng, GuangZhou). Each semen sample was directed into a sterile plastic cup and liquefied in an incubator at 37 °C (JingHong, ShangHai). After complete liquefaction of semen, 10 ul of semen sample was taken and dropped into the detection well of the sperm counting plate (SAS, BeiJing). A computer-aided semen analyzer (Suijia, Beijing, SSA) was used to detect sperm concentration, forward movement, and vitality. The SSA automatic detection system captured the movement trajectory of sperm through microscopic camera technology, and further analyzed the kinematic parameters and vitality. Kinematic parameters were divided into motion speed parameters, motion mode parameters, and spatial displacement degree parameters. Sperm motility was classified into forward movement, non forward movement, and inactivity based on their movement status. Obtain ≥5 visual fields and ≥200 sperm from each semen sample for observation and counting of the concentration, total number, and percentage of viable sperm at all levels. The test was performed using a constant temperature console (TC-01H, China). Fresh liquefied semen smears were air dried and analyzed for sperm morphology *via* Diff-Quik stained method (AnHui, Ankebio, Moscow, Russia). Only sperm with normal head (neck) and tail (middle and main segments) were considered morphologically normal, and all sperm in critical states were classified as abnormally shaped. For each semen sample, at least 200 sperm samples were analyzed using a double-blind method.

## Sperm DFI detection

The sperm chromatin structure assay (SCSA) was a reliable and most commonly used assay for the determination of DFI. The flow cytometry-based SCSA method was used to determine the DFI values. A sperm nuclear integrity staining kit (Zhejiang Xingbo

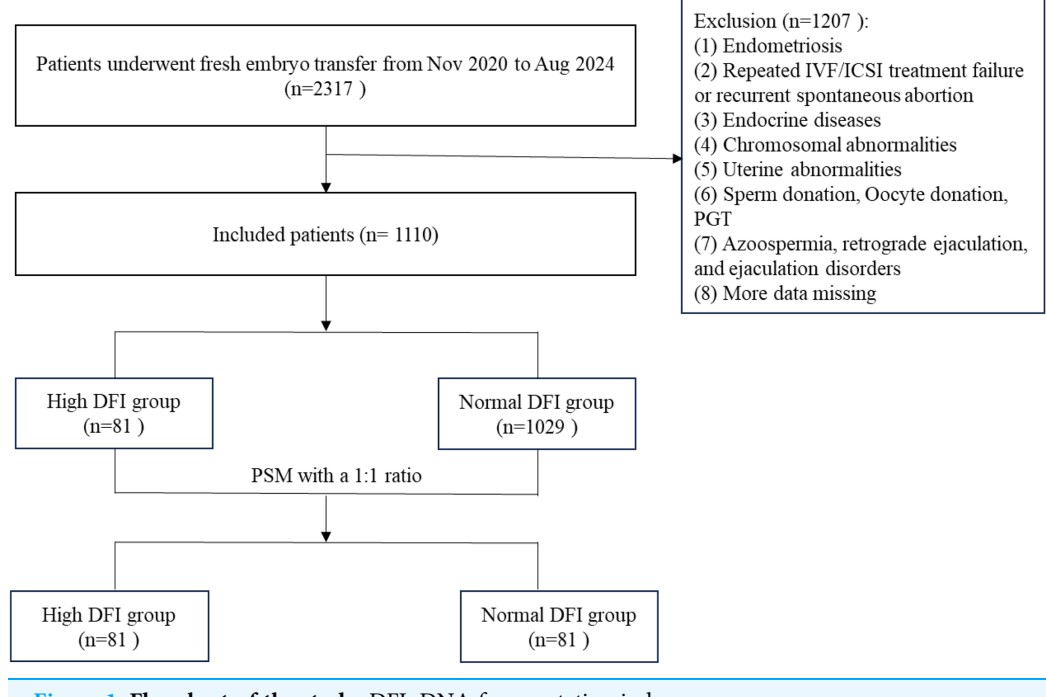

**Figure 1 Flowchart of the study.** DFI, DNA fragmentation index.

Biotechnology, Zhejiang, China) was used for testing, following the instructions in the kit manual. According to the instructions of the reagent kit, the sperm concentration was adjusted to $1 \times 10^6$–$2 \times 10^6$/mL, and 100 μL was used to test DFI. The chromatin in the damaged sperm nucleus forms a single chain after acid treatment, which binds with acridine orange and emits red or yellow fluorescence. The chromatin in normal sperm nuclei can maintain its intact double stranded structure after acid treatment and emit green fluorescence when combined with acridine orange. Fluorescence was detected by flow cytometry (Beckman Coulter, Brea, CA, USA) after the samples were stained with acridine orange. DFI (%) = green fluorescent sperm count/(green fluorescent sperm count + red fluorescent sperm count) × 100%.

## Sperm optimization for IVF/ICSI

The male partner abstained from sexual activity for 2–7 days and collected semen through masturbation on the day of oocyte retrieval. After the semen was completely liquefied, the sperm was selected by discontinuous density gradient centrifugation combined with the swimming-up method, as follows. First, sperm was prepared using a processing solution, including density gradient solution and swimming-up solution. The density gradient liquid was prepared using Vitrolife Ivrine Isolate (Vitrolife, Gothenburg, Sweden), with 0.5 ml of lower layer liquid and 0.5 ml of upper layer liquid loaded into a 15 ml Falcon centrifuge tube (Corning Incorporated, Corning, NY, USA) with a pointed bottom. The swimming-up solution used was Vitrolife G-IVF plus (Vitrolife, Gothenburg, Sweden),

2 ml was divided into 2 Falcon 2003 test tubes and equilibrated overnight in a 37 °C, 6% $CO_2$ incubator for future use. Then, (1) discontinuous density gradient centrifugation. Slowly add 2.0 ml of semen to the top layer of density gradient solution. After centrifugation at 300–400×g for 15 min, the supernatant was aspirated and discarded, leaving behind approximately 0.5 ml of sperm at the bottom. Add 1 ml of swimming-up solution to the sperm and mix thoroughly. Centrifuge again at 300–400×g for 5 min, then discard the supernatant and leave the bottom sperm at approximately 0.5 ml. (2) Sperm swimming-up. Add 0.5 ml of swimming-up solution to the sperm. The Falcon 1006 centrifuge tube was tilted at 45 degree angle in a 37 °C incubator with 5% $CO_2$ and saturated humidity. The process of sperm swimming-up lasted for 15 min, and then the upper sperm suspension was aspirated into another clean Falcon 1006 centrifuge tube for later use.

## Controlled ovarian stimulation and oocyte collection

In accordance with the patient's physical condition, controlled ovarian stimulation (COS) is performed *via* the ultralong GnRH agonist protocol, the long GnRH agonist protocol, the GnRH antagonist protocol or other protocols. Transvaginal ultrasound and serum hormone level testing (such as estradiol ($E_2$), luteinizing hormone (LH), follicle stimulating hormone (FSH), and progesterone (P)) are used to monitor follicle growth during the menstrual cycle. When the diameter of the dominant follicle is ≥19 mm or at least 3 follicles have a diameter ≥17.5 mm, a subcutaneous injection of 250 μg of human chorionic gonadotropin (HCG) should be given that evening, and oocyte aspiration should be performed after 36–40 h. Oocytes were cultured at 37 °C in a 5% $CO_2$ incubator for future use.

## Fresh IVF/ICSI-ET

The fertilization method was as follows: (1) IVF fertilization: The oocytes were cultured for 3-4 h before fertilization, with 50,000-100,000 sperm added to each oocyte. After 4–6 h, the cumulus cells were removed, and 16–18 h after fertilization the second polar body (2 pb) was observed. (2) ICSI fertilization: ICSI is performed 2–3 h after oocyte retrieval. Embryo culture and scoring were conducted according to standard operating procedures. The embryo scoring criteria are mainly based on morphological indicators. On the third day of 2PN fertilization, the embryos underwent cleavage, with 7–10 blastomeres of uniform size and a fragmentation rate of 0–20%, which were classified as high-quality cleavage embryos. The evaluation of blastocysts is based on three aspects: the degree of expansion of the blastocyst cavity, the number of inner cell mass, and the number of trophoblast cells. Fresh D3/D4/D5/D6 embryos can be transferred, and the number of embryos transferred varies from one to two according to the doctor's advice and the patient's requirements. Fourteen days after embryo transfer, a positive HCG blood test (HCG ≥ 50 mIU/ml) indicates a biochemical pregnancy. Five weeks after transplantation, B-ultrasound examination revealed a gestational sac and primitive pulsation in the uterine cavity, which confirmed clinical pregnancy.

## Outcome measures

MII oocyte rate = (number of MII oocytes/total number of retrieved oocytes) × 100%;
IVF 2PN fertilization rate = (2PN number/number of retrieved oocytes) × 100%;
ICSI 2PN fertilization rate = (2PN number/MII oocyte number) × 100%;

D3 high-quality embryos = sourced from 2PN fertilization, with 7–10 symmetrical blastomere cells, rated as grade I or II; D3 high-quality embryo rate = (number of D3 high-quality embryos/number of 2PN cleavages) × 100%;

Biochemical pregnancy rate = (number of biochemical pregnancy cycles/number of fresh transplant cycles) × 100%;

Clinical pregnancy rate = (number of clinical pregnancy cycles/number of fresh transplant cycles) × 100%;

Miscarriage rate = (number of miscarriage cycles/number of fresh clinical pregnancy cycles) × 100%;

Delivery rate = (number of delivery cycles/number of fresh transplant cycles) × 100%.

## Data statistics and analysis

R 3.3.4 was used for statistical analysis of the data. The Shapiro–Wilk test was used to test the normality of the measurement data. Normally distributed data are presented as the means ± standard deviations (X ± S), and intergroup comparisons were conducted *via* t tests. Quantitative data that conform to a skewed distribution or approximate normal distribution are represented by medians (M (Q1, Q3)), and intergroup comparisons were performed *via* the Mann Whitney U test. Count data are expressed as percentages (%) and were compared between groups *via* the chi square test or Fisher's test. The bilateral test method was used, and $p < 0.05$ was considered statistically significant.

## RESULTS AND ANALYSIS

### Comparison of semen parameters among different sperm DFI groups

After PSM, this study included 162 male patients with a median age of 33 years. The median sperm DFI was 29.79%. The median semen volume was 2.90 ml. The median sperm concentration was $48.98 \times 10^6$/ml. The median sperm motility was 40.15%. The forward movement of the sperm was 29.34%. The percentage of normal morphology sperm was 4.46%. According to the DFI value of sperm, there were 81 patients (50%) with a DFI ≥ 30% and 81 patients (50%) with a DFI < 30%. A comparison of the two groups revealed that sperm forward movement ($p < 0.001$), sperm motility ($p < 0.001$), and the percentage of sperm with normal morphology ($p = 0.001$) were significantly greater in the normal DFI group than in the high DFI group. However, there were no significant differences in male age ($p = 0.574$), abstinence days ($p = 0.244$), semen volume ($p = 0.523$), or sperm concentration ($p = 0.056$) between the two groups (Table 1).

### Correlation analysis between the sperm DFI and semen parameters

The correlations between the sperm DFI and routine semen parameters, as well as the associations between routine semen parameters and the DFI, are worthy of further exploration. Spearman correlation analysis revealed a negative correlation between the

**Table 1 Comparison of semen parameters between different sperm DFI groups.**

|  | Overall ($n$ = 162) | DFI < 30% ($n$ = 81) | DFI ≥ 30% ($n$ = 81) | $p$ |
|---|---|---|---|---|
| Male age (years) | 33.00 [30.00, 36.75] | 33.0 [30.0; 36.0] | 33.0 [31.0; 37.0] | 0.574 |
| Abstinence days | 4.00 [3.00, 4.00] | 4.00 [4.00; 5.00] | 4.00 [3.00; 4.00] | 0.244 |
| Semen volume (ml) | 2.90 [2.00, 3.95] | 2.90 [2.10; 4.00] | 3.00 [2.00; 3.70] | 0.523 |
| Sperm concentration ($10^6$/ml) | 48.98 [25.02, 91.96] | 55.8 [29.9; 93.6] | 37.6 [21.9; 81.2] | 0.056 |
| Sperm forward movement (%) | 29.34 [17.86, 41.55] | 37.0 [27.3; 46.5] | 22.3 [13.5; 31.5] | 2.06e−07 |
| Sperm motility (%) | 40.15 [21.65, 53.78] | 48.1 [35.9; 57.5] | 27.5 [18.4; 41.0] | 3.37e−07 |
| DFI (%) | 29.79 [11.72, 36.56] | 11.7 [9.05; 19.3] | 36.6 [33.6; 46.5] | <2.2e−16 |
| Normal morphology sperm (%) | 4.46 [4.07, 5.39] | 4.83 [4.31; 5.85] | 4.35 [3.83; 4.88] | 0.001 |

**Note:**
DFI, DNA fragmentation index.

sperm DFI and sperm motility (r = −0.44, $p$ = 3.32e−08), forward movement of sperm (r = −0.46, $p$ = 3.25e−09), and normal morphology of sperm (r = −0.25, $p$ = 0.000), of course, these linear correlations were not strong. However, there was no correlation between the sperm DFI and male age (r = 0.08, $p$ = 0.31), semen volume (r = −0.15, $p$ = 0.05), sperm concentration (r = −0.16, $p$ = 0.32), or BMI (r = 0.02, $p$ = 0.98) (Table 2 and Fig. 2).

## Comparison of pregnancy outcomes between the different sperm DFI group

After conducting PSM, there were no significant differences in terms of infertility duration ($p$ = 1.000), female age ($p$ = 0.615), female BMI ($p$ = 0.671), female basal FSH ($p$ = 0.879), female basal $E_2$ ($p$ = 0.967), female basal LH ($p$ = 0.683), female AMH ($p$ = 0.875), male BMI ($p$ = 0.693), COS protocol ($p$ = 1.000), number of oocytes retrieved ($p$ = 0.861), fertilization ($p$ = 0.962), or number of embryos transferred ($p$ = 0.747) between the two groups (Table 3). As shown in Tables 4 and 5, when the clinical outcomes of the two groups were compared, there were no significant differences in the 2PN fertilization rate (64.98% *vs.* 67.18%, $p$ = 0.362), D3 high-quality embryo rate (28.34% *vs.* 23.91%, $p$ = 0.107), biochemical pregnancy rate (71.60% *vs.* 71.60%, $p$ = 1), clinical pregnancy rate (65.00% *vs.* 65.00%, $p$ = 1), delivery rate (50.72% *vs.* 48.44%, $p$ = 0.928), miscarriage rate (7.25% *vs.* 6.25%, $p$ = 1), or singleton birth weight (3,350 g *vs.* 3,200 g, $p$ = 0.599) between the two groups.

## DISCUSSION

This study used the PSM statistical method to strictly control confounding factors and the SCSA method (Qiu et al., 2020) to detect the sperm DFI, aiming to explore the importance of the sperm DFI in evaluating male fertility and predicting IVF/ICIS-ET outcomes. This retrospective study included 81 patients with high DFI and 81 patients with normal DFI. An investigation of semen testing and fresh IVF/ICIS-ET outcomes revealed a certain correlation between the DFI and semen parameters, whereas the DFI was not correlated with clinical outcomes.

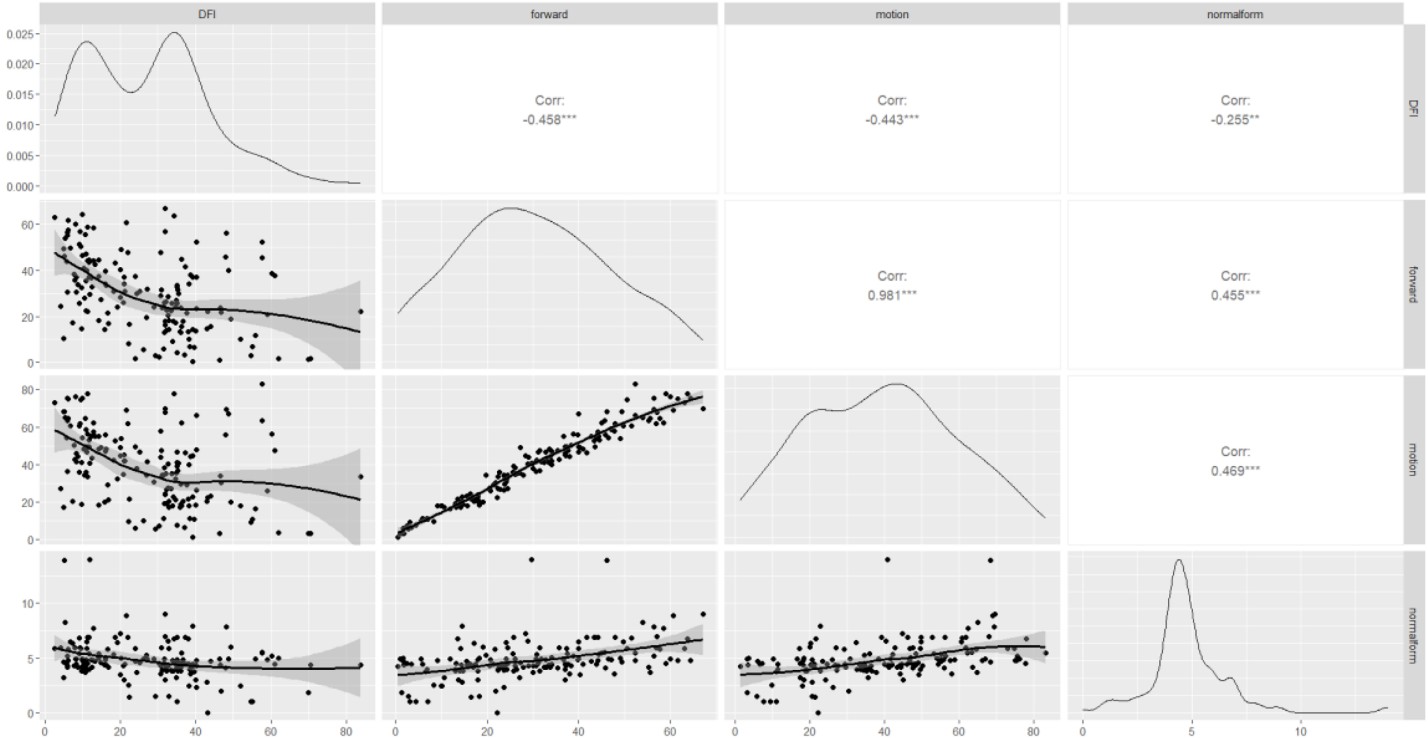

**Figure 2 Scatterplot matrix of correlations among DFI, sperm motility, forward movement, and normal morphology.** DFI, DNA fragmentation index; forward, sperm forward movement; motion, sperm motility; normal form, normal morphology sperm. Corr, correlation; ***moderate correlation; **weak correlation.                                    

Our study provided clinically relevant insights into the controversial role of sperm DFI in male fertility assessment. While confirming the weak association between DFI and conventional semen parameters (motility, forward movement, and normal morphology), the most significant clinical implication was that elevated DFI did not compromise fresh IVF/ICSI outcomes in tubal-factor infertility couples. This finding challenged the routine clinical practice of DFI testing in this specific population and warranted reconsideration of current guidelines.

Many studies have shown that the sperm DFI is closely related to semen parameters. This study revealed a negative correlation between sperm DFI and sperm motility, forward movement, and normal morphology in 162 fresh IVF/ICSI-ET cycles. Of course, these linear correlations were not strong. However, no correlation was detected between the sperm DFI and male age, BMI, or sperm concentration. Many studies support a negative but not strong linear correlation between the sperm DFI and sperm motility, forward movement, and normal morphology (*Du & Tuo, 2023*; *Liu et al., 2023*; *Zhang et al., 2021*). However, research results on the relationships between the sperm DFI and male age, BMI, and sperm concentration are inconsistent. *Du & Tuo (2023)* reported positive correlations between sperm DFI and age and BMI and negative correlations between sperm DFI and sperm concentration. *Luo et al.*'s *(2020)* study, *Yoshiakwa-Terada et al.*'s *(2024)* study and *Lu et al.*'s *(2020)* study revealed a positive correlation between sperm DFI and age, a negative correlation with sperm concentration, and no reported relationship with BMI

**Table 2 Spearman correlation analysis of the sperm DFI with semen parameters.**

|  | Male age | Semen volume | Sperm concentration | Sperm motility | Sperm forward movement | Normal morphology sperm | Male BMI |
|---|---|---|---|---|---|---|---|
| $r$ | 0.08 | −0.15 | −0.16 | −0.44 | −0.46 | −0.25 | 0.02 |
| $p$ | 0.31 | 0.05 | 0.32 | 3.32e−08 | 3.25e−09 | 0.000 | 0.98 |

Note:
    DFI, DNA fragmentation index; BMI, body mass index.

**Table 3 Comparison of baseline data between the two groups.**

|  | DFI < 30% ($n = 81$) | DFI ≥ 30% ($n = 81$) | $p$ |
|---|---|---|---|
| Infertility duration (year), $n$ (%) |  |  | 1.000 |
| ≤2 | 29 (35.8%) | 28 (34.6%) |  |
| >2 | 52 (64.2%) | 53 (65.4%) |  |
| Female age | 31.0 [29.0; 34.0] | 32.0 [28.0; 36.0] | 0.615 |
| Female BMI (kg/m$^2$) | 22.2 [20.0; 25.4] | 22.4 [20.5; 24.7] | 0.671 |
| Female basal FSH (IU/L) | 6.49 [4.94; 7.61] | 6.42 [4.76; 7.75] | 0.879 |
| Female basal E$_2$ (ng/L) | 35.6 [22.9; 56.2] | 37.0 [26.3; 47.9] | 0.967 |
| Female basal LH (IU/L) | 4.87 [2.89; 6.74] | 4.70 [3.25; 5.98] | 0.683 |
| Female AMH (ng/mL) | 3.00 [1.82; 4.25] | 2.91 [2.01; 4.01] | 0.875 |
| Male BMI | 24.2 [22.0; 26.2] | 24.4 [22.6; 26.0] | 0.693 |
| Protocol, $n$ (%) |  |  | 1.000 |
| Ultra long protocol | 75 (92.6%) | 75 (92.6%) |  |
| Antagonist protocol | 6 (7.41%) | 6 (7.41%) |  |
| Number of oocytes retrieved | 11.0 [7.00; 15.0] | 11.0 [8.00; 15.0] | 0.861 |
| Fertilization, % ($n$) |  |  | 0.962 |
| IVF | 47 (58.0%) | 47 (58.0%) |  |
| ICSI | 26 (32.1%) | 25 (30.9%) |  |
| R-ICSI | 8 (9.88%) | 9 (11.1%) |  |
| Number of embryos transferred, % ($n$) |  |  | 0.747 |
| 1 | 51 (63.0%) | 48 (59.3%) |  |
| 2 | 30 (37.0%) | 33 (40.7%) |  |

Note:
    DFI, DNA fragmentation index; BMI, body mass index; FSH, follicle stimulating hormone; E$_2$, estradiol; LH, luteinizing hormone; AMH, anti-Müllerian hormone; IVF, *in vitro* fertilization; ICSI, intracytoplasmic sperm injection; R-ICSI, rescue ICSI.

**Table 4 Comparison of embryo outcomes between the two groups.**

|  | DFI < 30% ($n = 81$) | DFI ≥ 30% ($n = 81$) | $p$ |
|---|---|---|---|
| Number of retrieved oocytes | 937 | 931 | – |
| Number of MII oocytes | 747 | 733 | – |
| 2PN fertilization rate (%) | 564/868 (64.98%) | 571/850 (67.18%) | 0.362 |
| D3 high-quality embryo rate (%) | 159/561 (28.34%) | 131/548 (23.91%) | 0.107 |

Note:
    DFI, DNA fragmentation index; 2PN, two-pronuclear.

**Table 5 Comparison of clinical outcomes between the two groups.**

| | DFI < 30% | DFI ≥ 30% | p |
|---|---|---|---|
| Biochemical pregnancy rate | 58/81 (71.60%) | 58/81 (71.60%) | 1 |
| Clinical pregnancy rate* | 52/80 (65.00%) | 52/80 (65.00%) | 1 |
| Delivery rate** | 35/69 (50.72%) | 31/64 (48.44%) | 0.928 |
| Miscarriage rate*** | 5/69 (7.25%) | 4/64 (6.25%) | 1 |
| Singleton birth weight (g)**** | 3,350 [3,050; 3,500], n = 30 | 3,200 [3,025; 3,500], n = 26 | 0.599 |

Notes:
* The follow-up period ended on September 18, 2024, and two patients did not reach the time for clinical pregnancy outcomes.
** The follow-up period ended on September 18, 2024. In the DFI < 30 group, 35 live births occurred, 10 cases did not reach the follow-up time, and two cases of ectopic pregnancy, which is 35/(81-12). In the DFI ≥ 30 group, 31 live births, one ectopic pregnancy, and 16 cases did not reach the follow-up time, which is 31/(81-17).
*** The follow-up period ended on September 18, 2024. The denominator is the same as the denominator of the delivery rate. There were five cases of miscarriage in the DFI < 30 group. There were four cases of miscarriage in the DFI ≥ 30 group.
**** The follow-up period ended on September 18, 2024. The DFI < 30 group included 30 cases of singleton live births. There were 26 singleton live births in the DFI ≥ 30 group.
DFI, DNA fragmentation index.

(*Lu et al., 2020*; *Luo et al., 2020*; *Yoshiakwa-Terada et al., 2024*). However, *Zhang et al. (2021)* reported that the sperm DFI was not correlated with sperm concentration. *Zhu et al. (2022)* reported a negative correlation between sperm DFI and BMI. *Liu et al. (2023)* reported a negative correlation between the sperm DFI and the sperm concentration, regardless of age, abstinence time, or normal morphology. The contradictory outcomes may be due to many potential confounding factors, such as abstinence time, infertile or fertile men or volunteers, different testing methods, *etc*. This study used the PSM statistical method to match the BMI and age of the males between the two groups so that there was no difference in the BMI of the males between the two groups. This may also be due to the similar ages of both spouses, as there was no difference in male age. Owing to the strict inclusion criteria, the sample size of this study was limited, and larger sample sizes are needed in the future to further research the relationship between male BMI and the sperm DFI.

The debate over whether the sperm DFI affects the clinical outcomes of embryo transfer has been ongoing. This study included 162 pairs of fresh IVF/ICSI-ET patients, who were divided into a high DFI group (DFI ≥ 30%) and a normal DFI group (DFI < 30%), to investigate the effect of the sperm DFI on embryo transfer. The infertility duration, female age, female BMI, female basal hormone levels, female AMH, male BMI, COS protocol, number of retrieved oocytes, fertilization, and number of transplanted embryos were similar between the two groups, reducing the influence of other factors on clinical outcomes. This study revealed that there were no statistically significant differences in the 2PN fertilization rate, D3 high-quality embryo rate, biochemical pregnancy rate, clinical pregnancy rate, delivery rate, miscarriage rate, or singleton birth weight between the two groups, which is consistent with previous research results (*Chen et al., 2020*; *Krog et al., 2024*; *Maghraby et al., 2024*; *Qi et al., 2022*). This may be related to the following reasons (*Liu et al., 2023*): (1) In embryo transfer, high-quality embryos are preferred, which may be the reason why the sperm DFI is not related to clinical pregnancy outcomes. (2) After

optimizing the sperm through the upstream method and density gradient centrifugation, the sperm with poor viability were eliminated. ICSI insemination results in the selection of sperm with good morphology and strong vitality, whereas IVF insemination with the zona pellucida can prevent the production of sperm with poor vitality. These factors weaken the impact of high-DFI sperm on clinical pregnancy outcomes (*Mantravadi & Rao, 2024*). (3) Early embryonic development is mainly controlled by maternal genes, with partial repair of sperm DNA damage after a period of successful pregnancy, activation of late-stage genes by the father, and high-sperm DFI embryo patients may not be able to continue pregnancy until 3 months or give birth. (4) Existing methods for detecting the sperm DFI can provide only the proportion of DFI-positive sperm at the cellular level and cannot quantitatively detect breakpoints at the DNA molecular level. DNA breakpoints show better correlation and predictive efficiency with assisted reproduction than DFI detected by the SCSA (*Yan et al., 2023*; *Zhou et al., 2024*). (5) Lifestyle factors such as smoking has been reported to be one of the important cause of sperm DNA fragmentation. When linking sperm DNA breakage to sperm quality and *in vitro* fertilization results, this factor may not have been taken into account.

Some studies have also produced different results. *Ebrahimi et al. (2024)* reported that a DFI > 15% is a risk factor for clinical pregnancy (OR = 0.27, $p = 0.011$). *Ibis et al. (2024)* reported that the DFI is a significant predictor of live birth outcomes (OR = 1.04; 95% CI [1.01–1.08], $p = 0.009$). *Zhang et al. (2023)* reported that an abnormally elevated DFI can reduce the clinical outcome of IVF-ET but does not affect the clinical outcome of ICSI-ET. *Wang et al. (2023)* reported a negative correlation between the DFI and the number of high-quality embryos with D3 (r = −0.347, $p < 0.001$) and the live birth rate (r = −0.185, $p = 0.028$). *Wang et al. (2022)* reported that for PCOS patients undergoing *in vitro* fertilization, fertilization with sperm with a relatively high DFI resulted in relatively low rates of high-quality blastocyst formation. The main reason for the differences in these research results is the use of different inclusion criteria, such as repeated failures in the literature, female obesity or PCOS, and male testicular sperm retrieval. This leads to a complex background of infertility in patients and a decrease in the ability of the mother to repair damaged sperm genes. This study included female infertility caused by fallopian tube factors, whereas male infertility was excluded because of azoospermia. The background of infertility was simple.

This study had several limitations. First, as the investigation focused specifically on tubal-factor infertility populations, our findings should not be extrapolated to cases of severe male factor infertility, recurrent pregnancy loss, or non-ART treatments. Second, the relatively small sample size constrained our ability to perform stratified analyses. Future studies with larger cohorts should conduct stratification based on: (1) female infertility etiology, (2) semen parameter thresholds, and (3) previous ART failure history. Third, the current study only examined sperm DFI; future research should explore advanced sperm assessment technologies, including quantitative DNA breakpoint detection and sperm chromatin maturity assays.

In summary, our current research suggests that the sperm DFI is weakly related to sperm motility, forward movement, and normal sperm morphology, and the correlations

are not strong. However, there is no correlation between the sperm DFI and the clinical outcomes of fresh IVF/ICSI-ET, and until now, there has been insufficient evidence to recommend the routine use of sperm DNA testing in evaluating the treatment of infertile couples.

### Funding

This work was supported by the Scientific Research Project of Jiangxi Provincial Health Commission (No. 202211108) and the Science and Technology Plan of Jiangxi Provincial Administration of Traditional Chinese Medicine (No. 2022B272). The funders had no role in study design, data collection and analysis, decision to publish, or preparation of the manuscript.

### Grant Disclosures

The following grant information was disclosed by the authors:
Scientific Research Project of Jiangxi Provincial Health Commission: 202211108.
Science and Technology Plan of Jiangxi Provincial Administration of Traditional Chinese Medicine: 2022B272.

### Competing Interests

The authors declare that they have no competing interests.

### Author Contributions

- Xiao Ju Wan conceived and designed the experiments, performed the experiments, analyzed the data, prepared figures and/or tables, authored or reviewed drafts of the article, and approved the final draft.
- Meizhen Huang conceived and designed the experiments, performed the experiments, analyzed the data, prepared figures and/or tables, authored or reviewed drafts of the article, and approved the final draft.
- Min Yu conceived and designed the experiments, performed the experiments, analyzed the data, prepared figures and/or tables, authored or reviewed drafts of the article, and approved the final draft.
- Tao Ding conceived and designed the experiments, performed the experiments, analyzed the data, prepared figures and/or tables, authored or reviewed drafts of the article, and approved the final draft.
- Zhihui Huang conceived and designed the experiments, performed the experiments, analyzed the data, prepared figures and/or tables, authored or reviewed drafts of the article, and approved the final draft.
- Zhiqin Zhang conceived and designed the experiments, performed the experiments, analyzed the data, prepared figures and/or tables, authored or reviewed drafts of the article, and approved the final draft.

- Xingwu Wu conceived and designed the experiments, performed the experiments, analyzed the data, prepared figures and/or tables, authored or reviewed drafts of the article, and approved the final draft.
- Jun Tan conceived and designed the experiments, performed the experiments, analyzed the data, prepared figures and/or tables, authored or reviewed drafts of the article, and approved the final draft.

## Human Ethics

The following information was supplied relating to ethical approvals (*i.e.*, approving body and any reference numbers):

The studies involving human participants were reviewed and approved by the hospital Ethics Committee The Reproductive Medicine Ethics Committee of Jiangxi Maternal and Child Health Hospital (SZYY-202407).

## Data Availability

The raw data are available in the Supplemental File.

## Supplemental Information

Supplemental information for this article can be found online at http://dx.doi.org/10.7717/peerj.19451#supplemental-information.

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
