# Peer review of "Correlation of the sperm DNA fragmentation index with semen parameters and its impact on fresh embryo transfer outcomes—a retrospective study"

_PeerJ, doi:10.7717/peerj.19451_

## Round 0.1 · original submission · Major Revisions

Please respond in detail to the comments of the reviewers

·

Basic reporting

Requires language editing. Avoid using terminologies which are not standard such as sperm supply cycle, oocyte supply cycle etc.. Instead cycles with donor sperm and oocytes may be mentioned

Experimental design

Inclusion criteria is not clear. Authors have included patients with tubal factor infertility and their one of the exclusion criteria was anatomical abnormalities that includes tubal factors. Please specify the exact conditions that the authors have excluded.
Lifestyle factors such as smoking has been reported to be one of the important cause of sperm DNA fragmentation. Authors should justify why they have not considered this factor while associating sperm DNA fragmentation on sperm quality and IVF outcomes
Methodology should be elaborated on semen analysis, density gradient preparation. Upstream word is not standard while explaining sperm wash method. Please replace with swim up.
Sperm morphology assessment using Pasteur staining method is not a WHO recommended test. Hence it is recommended to give the methodology with an appropriate citation.
Authors have followed short insemination IVF protocol where sperm is incubated with Oocyte cumulus complex for 4-6 hrs. However, it is impossible to see 2pn and 2pb at that time. Please mention the correct timing of fertilization check
Line # 99, please replace inner cell clusters with inner cell mass
Please mention the level of beta hCG for positive biochemical pregnancy

Validity of the findings

Authors have performed two group comparisons. However, they have used Kruskal test which is not appropriate test for two group comparison. Please justify.
When skewed distribution is present nonparametric test should be used like Mann Whitney U test? Please rerun the statistical analysis for an appropriate interpretation of the result.
Table 1: very small difference in sperm morphology (less than 0.5) has also been found to be statistically significant. Please recheck the analysis
Table 2: r=-0.25 is a very weak correlation. Please interpret accordingly

When authors have used PSM matching the female age, male body mass
index (BMI), female BMI, female infertility duration, female anti-Mullerian hormone (AMH), fertilization method, controlled ovarian stimulation (COS) protocol, and number of embryos transferred, what is the purpose of Table 3

Reviewer 2 ·

Basic reporting

The manuscript is written in clear and professional language, facilitating easy understanding of the study.

Relevant literature is cited, providing a robust context and background for the research question.

Figures and tables are well-organized and effectively illustrate the data and findings.

The abstract is concise and successfully captures the main findings and implications of the study.

Experimental design

The study design, which employs retrospective propensity score matching, is well-suited to address the research question while mitigating the impact of confounding variables.

Methods are thoroughly described, particularly for semen analysis and statistical techniques, ensuring that the study can be replicated by other researchers.

Validity of the findings

The conclusion, asserting no correlation between sperm DNA fragmentation index (DFI) and clinical outcomes, aligns with the data presented in the results section, indicating consistency and logical reasoning.

Additional comments

The manuscript demonstrates a solid foundation and contributes valuable insights to the field. Addressing the above criticisms would further enhance its impact and clarity.

Reviewer 3 ·

Basic reporting

They found that sperm DFI is related to sperm motility, forward movement, and normal sperm morphology and is an important indicator for evaluating semen quality. However, there is no correlation between the sperm DFI and the clinical outcomes of fresh IVF/ICSI-ET. There is no problem with the English language and references. If the tables were presented as graphs, their clarity could be improved.

Experimental design

If the tables were presented as graphs, their clarity could be improved. The methods of sperm motility and sperm DNA damage tests should be explained in more detail.

Validity of the findings

The number of patients is quite sufficient. The importance of sperm motility in treatment and the reasons why sperm DNA damage does not affect IVF/ICSI rates should be emphasized more in the discussion section.

---

## Round 0.2 · Minor Revisions

Your submission still needs some final revisions. Please respond to the comments below

·

Basic reporting

1. Language editing is required for the sections where authors have elaborated the methodology part in reply to the reviewers comments.

2. Replace internal cell mass with inner cell mass.

3. Replace semen concentration with sperm concentration, semen motility with sperm motility, semen DFI with sperm DFI, semen morphology with sperm morphology throughout the manuscript

Experimental design

No comment.

Validity of the findings

No comment.

Additional comments

1. Minor suggestion to authors to modify the first half of the conclusion as they have modified the results and agree that only weak correlation/association exists between sperm characteristics and sperm DFI.

---

## Round 0.3 · Minor Revisions

Please address the remaining reviewer comments.

·

Basic reporting

Suggested authors to replace semen concentration with sperm concentration, semen motility with sperm motility, semen DFI with sperm DFI, semen morphology with sperm morphology. However authors have modified other way round. Please use correct terminology

Experimental design

No comment

Validity of the findings

No comment

---

## Round 0.4 · Minor Revisions

Dear authors,

I am sorry it comes to you at this stage, but there are minor issues that ought to be resolved before acceptance:

- additional visual representations such as figures, graphs or charts e.g. displaying the correlation of DFI with semen parameters could enhance understanding for readers, and I don't understand why not even an inclusion/exclusion flowchart has been included? Why are there no visual records of data?

- Clearly articulate primary and secondary hypotheses

- in the discussion, a deeper analysis of the clinical implications of the findings is suggested

- Consideration of limitations should be explicitly addressed. For example, the retrospective nature of the study could introduce bias, and the reliance on specific methods for DFI measurement could affect generalization.... and suggest avenues for future research based on the findings.

- do not forget to include the ref for WHO Semen analysis manual

- methods and materials should be majorly revised for completeness and transparency (what exact materials were used? ref #? models and city/ country of instruments, etc)

Thank you for your understanding.

---

## Round 0.5 · accepted · Accept

Dear authors,

Thank you and congratulations! I am now accepting your manuscript for publication.